# Identification of Signature Genes of Dilated Cardiomyopathy Using Integrated Bioinformatics Analysis

**DOI:** 10.3390/ijms24087339

**Published:** 2023-04-16

**Authors:** Zhimin Wu, Xu Wang, Hao Liang, Fangfang Liu, Yingxuan Li, Huaxing Zhang, Chunying Wang, Qiao Wang

**Affiliations:** 1Department of Pharmacy, Hebei Medical University, Shijiazhuang 050017, China; 2Core Facilities and Centers, Hebei Medical University, Shijiazhuang 050017, China

**Keywords:** dilated cardiomyopathy, microarray, RNA-Seq, DEGs, robust rank aggregation

## Abstract

Dilated cardiomyopathy (DCM) is characterized by left ventricular or biventricular enlargement with systolic dysfunction. To date, the underlying molecular mechanisms of dilated cardiomyopathy pathogenesis have not been fully elucidated, although some insights have been presented. In this study, we combined public database resources and a doxorubicin-induced DCM mouse model to explore the significant genes of DCM in full depth. We first retrieved six DCM-related microarray datasets from the GEO database using several keywords. Then we used the “LIMMA” (linear model for microarray data) R package to filter each microarray for differentially expressed genes (DEGs). Robust rank aggregation (RRA), an extremely robust rank aggregation method based on sequential statistics, was then used to integrate the results of the six microarray datasets to filter out the reliable differential genes. To further improve the reliability of our results, we established a doxorubicin-induced DCM model in C57BL/6N mice, using the “DESeq2” software package to identify DEGs in the sequencing data. We cross-validated the results of RRA analysis with those of animal experiments by taking intersections and identified three key differential genes (including *BEX1*, *RGCC* and *VSIG4*) associated with DCM as well as many important biological processes (extracellular matrix organisation, extracellular structural organisation, sulphur compound binding, and extracellular matrix structural components) and a signalling pathway (HIF-1 signalling pathway). In addition, we confirmed the significant effect of these three genes in DCM using binary logistic regression analysis. These findings will help us to better understand the pathogenesis of DCM and may be key targets for future clinical management.

## 1. Introduction

Dilated cardiomyopathy (DCM) is a non-ischaemic myocardial disease with functional and structural myocardial abnormalities that manifests clinically as left ventricular or biventricular dilatation and systolic dysfunction in the absence of hypertension, valvular disease, coronary artery disease or congenital heart disease [1]. The causes of this disease include a variety of common factors such as genetic susceptibility, inflammation, infection, toxin exposure, and abnormalities of the autoimmune system [2]. It can occur at any age but is more common in young male adults, 35% of whom often have genetic mutations [3,4]. Although scientists have identified a number of broad genetic variants encoding proteins associated with the cytoskeleton, nucleic acids and gene expression that may be involved in the disease process of DCM [5], the major drivers of DCM pathogenesis remain incompletely elucidated. DCM remains an important contributor to heart failure and sudden cardiac death and is a major indication for heart transplantation in the entire population. Therefore, further deciphering the underlying molecular mechanisms in the pathogenesis of DCM will facilitate the development of effective strategies to treat patients with cardiac dysfunction and prevent severe heart failure and sudden cardiac death.

In recent years, with the rapid development of high-throughput sequencing technologies such as gene chips and next-generation sequencing, we can obtain a large amount of data information in a relatively short period, which is often stored by researchers in large bioinformatics databases, such as Gene Expression Omnibus (GEO). Other researchers are free to download the information they need from these databases for secondary analysis studies. These conveniences create further opportunities to discover new mechanisms of disease occurrence and relevant therapeutic targets. 

Integrating datasets from multiple studies in publicly available databases to identify differentially expressed genes in disease states helps to understand the pathogenesis of various diseases, including DCM. Microarray-based gene expression profiling research has been widely used in the cardiac muscle tissue from DCM patients to identify biomarkers. However, different microarray studies may be based on different analysis platforms, contain different sample sizes and sample sources, and some data outliers may be scattered across these datasets. These inconsistencies pose significant challenges for the integrated analysis of datasets and may affect the accuracy of the results. 

Robust rank aggregation (RRA), a rank aggregation method based on sequential statistics, was first proposed by Kolde et al. in 2012 and is widely used in bioinformatics analysis [6]. It has many advantages over traditional analysis methods, such as strong stability against noise, relatively high calculation efficiency, assigning significance scores to the elements in the results, and sorting out incomplete rankings [6]. These superior data handling capabilities perfectly overcome inconsistencies between datasets and the output is given a *p*-value for each element to represent their importance. 

In this study, the RRA algorithm was used to integrate public database resources and combined with animal experiments to jointly identify DEGs in DCM. Multiple DCM-related microarray datasets in the GEO database were first searched by several keywords. The “LIMMA” R package was used to screen out the differentially expressed genes in each microarray, and then the RRA algorithm in bioinformatics was used to integrate the results of these analyses and screen out the more reliable DEGs. To further increase the reliability of our findings, we established a mouse model of doxorubicin-induced dilated cardiomyopathy. RNA-Seq analysis was performed on ventricular tissue from mice, and the “DESeq2” software package was used to identify DEGs in the sequencing data. The differential genes obtained from both dataset-based RRA analysis and animal experiment were intersected to obtain reliable specific expressions of genes associated with DCM. The significance of these genes for DCM was further confirmed by binary logistic regression analysis. In addition, we used the “ClusterProfiler” R package to perform GO and KEGG functional enrichment analysis on DEGs screened by both analysis strategies. We cross-tabulated the functional enrichment results to identify those biological processes and signalling pathways associated with DCM.

## 2. Results

### 2.1. Microarray Information and DEGs in Six Datasets

In this study, GSE3585, GSE29819, GSE42955, GSE43435, GSE79962, and GSE84796 were included, containing 55 DCM patients and 44 control patients. Information about the samples included in the study is presented in Appendix A. To eliminate individual differences between samples, normalization of each of the six datasets was performed using the RMA method in the Affymetrix toolkit. According to the pre-set cut-off criteria, the differential expression analysis was carried out using the “LIMMA” package in R software (version 3.50.3). Table 1 summarizes the details of the datasets included in this study. Figure 1 shows all the DEGs in the six datasets and tags the significantly changed genes.

### 2.2. RRA Integrated Analysis

The comprehensive analysis of multiple groups of data plays an important role in high-throughput data analysis. The RRA method has a good balance to the sample noise and gives a score for each element in the result. The lower the score of the RRA analysis results, the more reliable the screened differential genes are. A total of 195 differential genes were identified by RRA integration analysis, of which 88 were upregulated and 107 were downregulated in expression (Appendix A). 

### 2.3. Functional Annotation of RRA

One hundred and ninety-five differential genes were introduced into the R package “ClusterProfiler” (version 4.2.2) for GO and KEGG analysis. The analysis results showed that extracellular structure organisation, external encapsulating structure organisation, and extracellular matrix organisation were significantly enriched for biological processes. Collagen-containing extracellular matrix was significantly enriched for the cellular component. Extracellular matrix structural constituent and glycosaminoglycan binding was significantly enriched for molecular function (Figure 2). The results of the KEGG pathway enrichment analysis were shown in Figure 3.

### 2.4. DEGs in Animal Experiments

Echocardiography and related cardiac function index showed that doxorubicin successfully induced dilated cardiomyopathy (Figure 4). Quality control and sequencing information for the samples (Appendix A) showed that Q20 ranged from 97.36% to 97.76% and Q30 ranged from 92.32% to 93.34% for all samples, with an error rate of no more than 0.04%, indicating that the sequencing results were reliable. In the analysis of RNA-Seq, independent statistical hypothesis tests were performed on a huge number of genes to obtain DEGs. To avoid the problem of possible excessive false positives in the analysis results, we corrected the *p*-values obtained from the original statistical analysis when performing the analysis of variance. In the end, we screened 224 upregulated genes and 138 downregulated genes (Figure 5). The list of differential genes is presented in Appendix A. Combining the results of RNA-Seq analysis and RRA, we confirmed that *BEX1* and *RGCC* expression levels were upregulated and *VSIG4* expression levels were downregulated in DCM (Figure 6A,B).

### 2.5. Enrichment Analysis of RNA-Seq

According to the results of RNA-Seq analysis, we imported 362 differential genes into ClusterProfiler for GO analysis and KEGG analysis. Next, we sort by *p*-value and present the top 30 results in Appendix A. Finally, we intersect the results of RRA enrichment analysis with those of RNA-Seq enrichment analysis. The results showed that four biological processes were significantly enriched in both RRA and RNA-Seq assays, including extracellular matrix organisation (GO:0030198), extracellular structure organisation (GO:0043062), sulfur compound binding (GO:1901681), and extracellular matrix structural constituent (GO:0005201). Secondly, KEGG pathway enrichment analysis showed that the HIF-1 signalling pathway was significantly enriched in both analyses, with *NPPA*, *PIK3CA*, *PIK3R1*, and *STAT3* being human-derived aberrant genes in the pathway and *Eif4ebp1*, *Cdkn1a*, *Mknk2*, *Angpt1*, and *Tfrc* being mouse-derived aberrant genes in the pathway (Figure 6C,D).

### 2.6. Results of Binary Logistic Regression Statistical Analysis

The binary logistic regression is widely used in medical field research as it allows analysis of the correlation between different factors. As a very practical statistical method, binary logistic regression appears to be efficient and powerful, analysing the effect of each independent variable on binary outcomes by quantifying its unique contribution [13]. The expression profiles of the six previously obtained datasets were used to trace the expression levels of the three differential genes we identified. Then, the association analysis could be carried out by quantifying the contribution of these genes to DCM. The results of our output were shown in Table 2. The *p*-values for *BEX1*, *RGCC*, and *VSIG4* were 0.004, 0.049, and 0.030, respectively, showing significance at the horizontal level, so they all had a significant effect on DCM. This means that if *BEX1* is expressed at an upregulated level in an individual, the probability of DCM is increased by 353.838%. If *RGCC* was upregulated in an individual, the probability of DCM was increased by 173.106%. If *VSIG4* was downregulated in an individual, the probability of DCM was increased by 200.869%.

## 3. Discussion

Clinically, patients with DCM have varying degrees of ventricular dilatation and systolic dysfunction. DCM causes a range of cardiovascular diseases, including progressive heart failure [14]. So far, a variety of mechanisms have been reported, including altered metabolic profiles, transcriptional dysregulation, and so on [15]. At the same time, studies have shown that there is an interaction between susceptible polygenes and the environment in the pathogenesis of DCM [3]. Therefore, identifying the key susceptibility genes of DCM is of great significance for the further study of the pathogenesis of DCM and the exploration of effective treatments. The RRA algorithm has many advantages over traditional analysis methods, such as strong stability against noise, relatively high calculation efficiency, assigning significance scores to the elements in the results, and sorting out incomplete rankings [6]. Therefore, we used RRA to screen the public database for differential genes in microarray datasets. To further increase the reliability of the results, we established a DCM mouse model using doxorubicin and performed deep sequencing analysis. The results screened by the RRA algorithm were cross-validated with the results from animal experiments in an intersection-taking approach, resulting in the identification of DCM-related differential genes, important biological processes and signalling pathways.

Combining the results of RRA and RNA-Seq analysis, we confirmed that *BEX1* and *RGCC* expression levels were upregulated and *VSIG4* expression levels were downregulated in DCM patients. We also corroborated their significance for DCM using binary logistic regression analysis. The results of statistical analysis showed that they had a significant impact on the occurrence of DCM. It is worth noting that we also found that the gene ontology terms such as extracellular matrix organisation, extracellular structure organisation, sulphur compound binding and extracellular matrix structural constituent as well as the KEGG pathway HIF-1 signalling pathway may be highly related to the occurrence and development of DCM.

Brain-expressed X-linked 1 (*BEX1*) belongs to the BEX gene family, which consists of *BEX1*, *BEX2*, *BEX3*, *BEX4*, *Tceal1*, *Tceal3*, *Tceal5*, *Tceal6*, *Tceal7*, *Tceal8*, and *Tceal9* [16]. In agreement with our findings, AronowBJ et al. found that *BEX1* showed upregulated expression levels in the hearts of mice with heart failure [17]. Accornero et al. demonstrated that forced expression of *BEX1* in mouse hearts using transgenic means ultimately led to more severe cardiac dysfunction and cardiac hypertrophy, whereas *BEX1* knockout mice exhibited a more favourable cardiac functional state [18]. In addition, they demonstrated a correlation between *BEX1* and the expression of the cardiac pro-inflammatory factor TNF-α [18]. Thus, *BEX1* may be a novel intracellular immunomodulator that mediates cardiac dysfunction through chronic pathological injury. 

Regulator Of Cell Cycle (*RGCC*), also known as *RGC32*, is expressed mainly in placenta, liver, kidney, pancreas, skeletal muscle and aortic endothelial cells, and weakly in heart and brain [19]. Although *RGC32* is not expressed in the myocardium more than other tissues, it may play a non-negligible role in cardiac disease. RGC32 was shown to physically bind to AKT and be induced to activate by AKT phosphorylation, and activated RGC32 increased the expression levels of Rho (Ras homologous log gene family, member A) and ROCK (Rho-associated protein kinase 1) [20,21]. It has been well established that the Ras homolog gene family is a regulator of ROCK. Overexpression of RhoA causes ROCK recruitment, so what role ROCK signalling plays depends on the cell type downstream of Rho and the function it plays. Myosin regulatory light chain 2 (MLC) is one of the substrates of ROCK and the phosphorylated MLC is completely activated, which then drives the contraction of actin by activating the active site of the ATPase of myosin [22]. Activated actin rapidly generates contractile forces that produce myoconstriction and lead to intense cellular tightening, a process that drives nuclear breakdown, dynamic membrane blistering and the production of apoptotic vesicles. Several studies have revealed the role of MLC phosphorylation and cytoskeletal remodelling (actin–myosin) in driving these deleterious processes [23]. Cleavage of the C-terminal autoinhibitory structural domain of ROCK1 by caspase-3 leads to a dramatic activation of ROCK1, ultimately causing a dramatic increase in phosphorylation of MLC, which appears to be a key initiation program for apoptosis [24]. Activation of the Rho/ROCK axis is an important pathway in the pathogenesis of heart failure, and it plays a crucial role in promoting apoptotic signalling in ventricular remodelling and functional failure [25]. Notably, the deleterious role of RGC32 as an important upstream signalling node for Rho/ROCK axis activation in myocardial diseases such as dilated cardiomyopathy and heart failure is self-evident. Elsewhere, Li et al. demonstrated that plasma *RGC32* levels were elevated in DCM patients and that *RGC32* levels were negatively correlated with the Treg/Th17 ratio [26]. Th17 cells promote an immune response that may impair cardiac function through a fibrotic response [27]. In contrast, Treg cells have anti-inflammatory effects and have been shown to be protective against viral and autoimmune cardiomyopathies [28]. Thus, increased levels of *RGC32* expression may simultaneously disrupt Treg/Th17 homeostasis and thus exert detrimental effects on the heart (Figure 7). In summary, the role of *RGC32* as an important signalling node in myocardial disease cannot be ignored and it has the potential to become a key target for the treatment of myocardial disease in the future. 

VSIG4 is V Set and Ig domain-containing 4, also known as CRIg, and is a member of the B7 family. VSIG4 is mainly expressed in resting macrophages and human dendritic cells, where it plays a key role in the process of innate and adaptive immunity [29,30]. On the one hand, studies have reported that VSIG4 has an inhibitory effect on T cells and NKT cells in vivo, which in turn reduces those pro-inflammatory cytokines that are harmful to the body, such as IL-17A, TNF-α and IFN-γ [31]. IL-17 and IFN-γ are differentially involved in the recruitment of inflammatory cells in cardiomyocytes and the remodelling of myocardial tissue; TNF-α mediates apoptosis and impairment of myocardial function via the NF-κB pathway [32,33,34]. This suggests that *VSIG4* has the potential to exert a protective effect on the heart through the above pathways. On the other hand, macrophage-induced inflammatory responses have an important role in driving the pathogenesis of various diseases, including various cardiac diseases [35,36]. Excitingly, it has been demonstrated that VSIG4 restricts pyruvate metabolism in mitochondria during oxidative phosphorylation by activating PI3K/Akt-STAT3 signalling channels, thereby inhibiting the polarisation of macrophages in a pro-inflammatory direction [37]. This function of regulating macrophage differentiation is another solid evidence for the protective function of *VSIG4*. In addition to this, it was shown that injection of VSIG4-Ig fusion molecules into mice inhibited T cell-induced Th cell-dependent IgG immune responses and that this immunosuppressive effect may play an important role in cardioprotection [38]. The multiple evidence above suggests that *VSIG4* is involved in the protection of cardiac function through immunomodulatory effects, and therefore activation of *VSIG4* expression in the myocardium may be an important immunotherapy for the treatment of various cardiomyopathies, including DCM.

We found that gene ontologies extracellular matrix organisation, extracellular structure organisation, sulphur compound binding and extracellular matrix structural constituent may be involved in the occurrence and development of DCM. The extracellular matrix is found in all tissues and organs and is a dynamic and ever-changing mixture of non-cellular components. Its main components are various fibrous proteins, including elastin, collagen, laminin, fibronectin and glycoproteins [39]. During the normal operation of mammalian myocardium, the extracellular matrix plays a vital role. It can not only conduct important molecular signals in disease and health but can also provide mechanical support to the heart [40]. There is ample evidence to suggest a direct correlation between the expansion and contraction of the extracellular matrix and cardiac dysfunction. The metalloproteinase MMP8 in the cell matrix induces macrophages to differentiate into a pro-inflammatory phenotype, thereby indirectly recruiting chemotactic inflammatory cells to the cardiac interstitium causing myocardial dysfunction [41]. The myocardium contains collagen, which is the main determinant of the structural integrity and mechanical function of the myocardium. The changes of myocardial structure and function in patients with dilated cardiomyopathy may be caused by the degradation of collagen fibres or the decrease in matrix elasticity [42]. A study shows that there is a clear correlation between collagen degradation and ventricular remodelling. The activity of matrix metalloproteinases in Syrian hamsters with cardiomyopathy increased greatly, so that the collagen volume fraction was lower than the normal level, which was consistent with the changes in ventricular wall thinning and ventricular dilatation in cardiomyopathy [43]. In addition, other extracellular substrates such as laminin, osteopontin, thromboreactive protein, fibronectin, tendinin-C, and periosteum proteins play important roles in the signalling process between cells [44,45,46,47,48].

In our KEGG enrichment analysis, the HIF-1 signalling pathway was significantly enriched with aberrant human-derived genes including *NPPA*, *PIK3CA*, *PIK3R1*, and *STAT3*, and aberrant mouse-derived genes including *Eif4ebp1*, *Cdkn1a*, *Mknk2*, *Angpt1*, and *Tfrc*. Except for *STAT3*, *Angpt1*, and *Tfrc*, the expression levels of the other six genes were upregulated. Hypoxia-inducible factor (HIF)-1α is an important transcription factor that regulates hypoxic response genes and plays a unique role in the hypoxic response to low levels of oxygen in tissues and organs [49,50]. At normal oxygen levels, oxygen-dependent prolyl-4-hydroxylase (PHDs) hydroxylates the proline residue of the HIF-α subunit. The hydroxylated HIF-α binds to an E3 ubiquitin ligase (pVHL) and is recognised as a substrate for the pVHL complex, which in turn degrades the HIF protein. Additionally, the inhibitory factors of HIFs, FIHs, hydroxylate the amino acid disability of the HIF protein, which in turn inactivates HIF. Under hypoxic conditions, the hydroxylation activity of PHDs and FIHs is inhibited and HIF-α migrates to the nucleus to bind to HIF-1β to form a heterodimer, which binds to the hypoxia response element (HRE) and thereby activates various downstream signalling pathways [49,51,52,53]. Under conditions of acute hypoxia, stabilisation of HIF induces a series of adaptive responses that are protective of the tissues. HIF-2α induces activation of vascular endothelial growth factor (VEGF) and erythropoietin (EPO), both of which have a role in improving oxygen supply to areas of myocardial ischaemia [54]. Under acute hypoxic conditions, the accumulation of ATP/ADP in the cell stroma increases dramatically and HIF1A binds to the active region of CD73 thereby increasing the level of adenosine in the tissue [55]. Extracellular adenosine factors can bind to the corresponding adenosine receptors to activate the adenosine signalling pathway, and the adenosine-mediated signalling pathway can improve tolerance to hypoxia in acutely ischaemic and hypoxic tissues [55,56]. Therefore, this protective mechanism plays an important role in various agents of ischaemic heart disease, such as ischaemia-reperfusion injury and acute myocardial infarction (Figure 8) [57,58]. However, in conditions of chronic hypoxia in myocardial tissue such as dilated cardiomyopathy and heart failure, the long-term stabilisation of HIF may have a pernicious impact on the heart [59]. As we described, transgenic mice forcibly expressing HIF-1α spontaneously exhibited signs of heart failure such as thickened cardiac septa and reduced shortening fraction over time [60]. Li et al. Knockdown of VHL from mouse hearts caused accumulation of FIH, leading to myogenic fibrillar disorders, lipid accumulation and severe structural abnormalities in the nucleus, resulting in loss of myocardial function and deformation, and ultimately the development of heart failure [61]. Prolonged activation of HIF due to loss of PHD prolyl hydroxylase may lead to cardiac dysfunction through interstitial oedema, altered tissue structure or signalling between cardiac myocytes [62]. In addition, a drug called carvedilol exhibited inhibition of HIF-1α, which may be a key factor in its treatment of volume overload heart failure [63]. From the early stages of DCM onset to the development of severe heart failure, cardiomyocytes inevitably suffer from varying degrees of hypoxia due to impaired cardiac pumping function, and this hypoxic state is prolonged. HIF-1 may therefore be a key pathway of myocardial damage during the disease process of DCM, and the development of drugs targeting HIF-1 will contribute to the clinical management of DCM. 

In summary, using the very robust RRA algorithm, we screened a total of 196 DEGs in multiple microarray datasets, including 88 upregulated genes and 108 downregulated genes. In addition, we established a DCM mouse model using doxorubicin and performed deep sequencing analysis to identify 362 differential genes, including 224 upregulated genes and 138 downregulated genes. We performed functional enrichment analysis on the DEGs obtained from both methods separately. Ultimately, we cross-validated the results of the RRA analysis with the results of animal experiments by taking intersections and identified three key differential genes (including *BEX1*, *RGCC* and *VSIG4*) associated with DCM as well as many important biological processes (extracellular matrix organisation, extracellular structural organisation, sulphur compound binding, and extracellular matrix structural components) and a signalling pathway (HIF-1 signalling pathway). It is exciting to note that the genes and associated signalling pathways we have identified play a crucial role in the pathology of DCM and they may become important targets for future clinical treatment.

## 4. Materials and Methods

### 4.1. DCM Related Microarray Datasets

Expression profiles of DCM-related microarray datasets were obtained from Gene Expression Omnibus (GEO, www.ncbi.nlm.nih.gov/geo (accessed on 16 May 2022)) [64]. We systematically searched microarray studies using the following keywords: “Dilated cardiomyopathy”, “Gene expression”, “*Homo sapiens*” and “Microarray”. Samples of DCM patients and healthy subjects were screened for inclusion in the experiment. The gene expression data of 44 healthy samples and 55 DCM patients were used in this study. Raw data or gene expression profiles through arrays can be obtained in GEO.

### 4.2. Acquisition and Analysis of Microarray Datasets

First, we downloaded the gene expression matrix and corresponding annotation information for the microarray datasets from the GEO database. Probe IDs for microarrays were translated into gene symbols using the appropriate annotation files. If multiple probes corresponded to the same gene symbol, we retained the average expression value. To eliminate individual differences between samples, normalization of each of the six datasets was performed using the RMA method in the Affymetrix toolkit [65]. The “limma” R packet (http://www.bioconductor.org/packages/release/bioc/html/limma.html (accessed on 10 June 2022)) [66] was used to identify differentially expressed genes between ventricular tissues of DCM patients and normal left ventricular tissues in each microarray dataset. The *p*-value < 0.05 and |log2FC| > 0.5 were regarded as the cut-off criteria to determine DEGs.

### 4.3. Integration Analysis of RRA Algorithm

The use of the RRA (http://127.0.0.1:15432/library/RobustRankAggreg/html/aggregateRanks.html (accessed on 12 June 2022)) method to identify reliable DEGs minimized inconsistency and effectively integrated the results of multiple microarray studies [6]. The RRA algorithm assigns an adjusted *p*-value to each output element, with a smaller *p*-value representing a higher ranking of the gene in the final list. Prior to RRA analysis, we first generated upregulated and downregulated genes for each dataset from expression fold change values between DCM cases and control samples. The list of differential genes for the full datasets was then integrated using the ‘Robust Rank Aggregation’ R package. Genes with *p*-value < 0.05 and the |logFC| > 0.5 were considered significant genes.

### 4.4. Functional Enrichment Analysis of RRA Results

“ClusterProfiler” (http://www.bioconductor.org/packages/release/bioc/html/clusterProfiler.html (accessed on 25 June 2022)) R package is a tool used for statistical analysis and visualization of functional maps of genes and gene clusters. We imported the important genes exported from the RRA analysis for GO functional enrichment analysis and KEGG pathway analysis to investigate the potential functions of these genes.

### 4.5. DCM Animal Model Establishment

All animal experiments were approved by the animal ethics committee of Hebei Medical University (IACUC-Hebmu-2022001). The DCM mouse model was established by intraperitoneal injection of doxorubicin hydrochloride. Six male C57BL/6N mice were purchased from Beijing Vital River Laboratory Animal Technology Co., Ltd. (Beijing, China). Mice are housed at 25 °C, kept on a 12-h light/dark cycle and have free access to water and food. The mice were randomly divided into two groups: Control (*n* = 3) and DCM (*n* = 3). The mice in the DCM group were injected intraperitoneally with doxorubicin hydrochloride (Haizheng Pfizer Pharmaceutical Co., Ltd., Hangzhou, China) at a dose of 18 mg/kg (2 times/week for 2 weeks, DCM group). Control mice were given an equal volume of saline in the same way. After one week, the Vevo^®^ 2100 system (Visual Sonics, Inc., Toronto, Ontario, Canada) equipped with a 30-MHz transducer was used to assess the cardiac status of the above two groups of mice and to determine the success of the model. Finally, heart tissue was collected for follow-up analysis after 8 h of fasting.

### 4.6. RNA-Seq and Data Analysis

Total RNA was extracted from mouse heart tissue using TRIzol^®^ reagent. All operations were in accordance with the manufacturer’s instructions (Magen Biotech Co., Ltd., Shanghai, China). The extracted RNA samples were assayed using the Nanodrop ND-2000 system (Thermo Fisher Scientific, Wilmington, DE, USA) with an absorbance of A260/A280. The RIN value of the extracted RNA was determined using the Agilent Bioanalyzer 4150 system (Agilent Technologies, Palo Alto, CA, USA). Only qualified RNA samples can be used to build the library. Paired-end libraries were prepared using the ABclonal mRNA-Seq Lib Prep Kit (ABclonal Technology Co., Ltd., Wuhan, China) and all procedures were carried out according to the kit instructions. The PCR products were purified using the AMPure XP system (Beckman Coulter, Bria, CA, USA), and the quality of the library was evaluated using the Agilent Bioanalyzer 4150 system. Finally, the prepared samples were sequenced on MGISEQ-T7 to produce 150 bp paired-end readings. Processing of raw read data in fastq format using perl scripts. When processing the read data, the number of lines whose cross-region quality value is less than or equal to 25, accounts for more than 60% of the measured values of the entire read, and the reads whose basic information ratio cannot be determined to be greater than 5% are also eliminated together. Clean reads were then matched to the reference genome using HISAT2 version 2.2.1 software (http://daehwankimlab.github.io/hisat2/ (accessed on 10 April 2022)) to obtain mapped reads. Feature counts (http://subread.sourceforge.net/ (accessed on 12 April 2022)) were used to calculate the number of reads for each gene. The FPKM for each gene was then calculated based on the length of the gene and the count corresponding to that gene was read. The data have been deposited in NCBI’s Gene Expression Omnibus and are accessible through GEO Series accession number GSE224211 (https://www.ncbi.nlm.nih.gov/geo/query/acc.cgi?acc=GSE224211 (accessed on 11 April 2023)). Differentially expressed genes were screened using the DESeq2 package (http://www.bioconductor.org/packages/release/bioc/html/DESeq2.html (accessed on 1 May 2022)) and genes with |log2FC| > 1 and Padj < 0.05 were considered to be significantly changed. The differential genes from the analysis were intersected with the results of the RRA analysis.

### 4.7. Enrichment Analysis of RNA-Seq Results

ClusterProfiler R software package was used to analyse the GO function enrichment and KEGG pathway enrichment of the differential genes obtained from RNA-Seq analysis. When *p* < 0.05, it is considered that the GO or KEGG function is significantly enriched. To enhance the reliability of the results, we crossed the results of the RNA-Seq enrichment analysis with the previous RRA results to obtain GO and KEGG that showed abnormalities in both humans and mice. For all cross-validation we used the online analysis tool Venny (https://bioinfogp.cnb.csic.es/tools/venny/index.html (accessed on 24 March 2023)) to make Venn diagrams.

### 4.8. Binary Logistic Regression Analysis

To confirm the significance of the DEGs we obtained, we used a statistical approach of binary logistic regression to analyse the correlation between DEGs and DCM [13]. Based on the gene expression profiles of the six previously obtained datasets, the expression levels of target differential genes in each group of clinical samples were screened (including control and DCM). The average expression value of the target gene in each dataset was used as a cut-off to distinguish the high and low expression levels of the target gene in different samples. According to the changing trend of our cross-validated genes in patients, the positive expression is defined as “1” and the negative expression is “0”. The sample data from the six collated datasets were combined and imported into the SPSS online analysis website (https://www.spsspro.com/ (accessed on 24 March 2023)) for binary logistic regression analysis. The significance of the OR value for the dummy variables 0–1 categorical independent variable is that the categorical level changed from 0 to 1 and the probability of an experimental group event occurring changed by (OR value − 1)% over the probability of a control group event occurring.

## Figures and Tables

**Figure 1 ijms-24-07339-f001:**
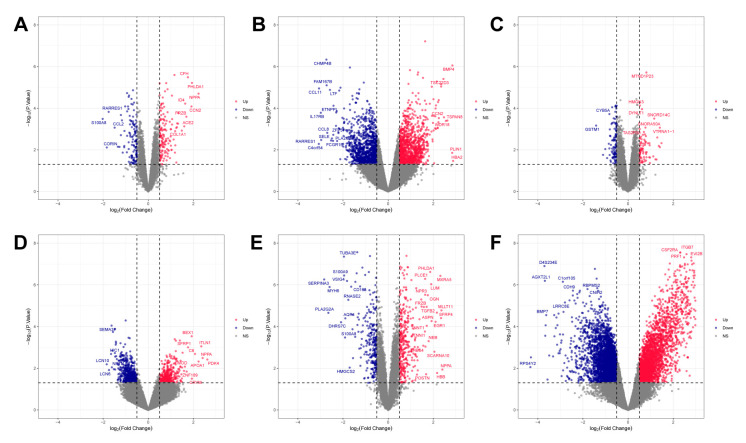
Volcano plots of the DEGs for the six datasets, where red dots represent upregulated genes, blue dots represent downregulated genes, and grey dots represent genes that did not change significantly. (**A**) GSE3585, (**B**) GSE29819, (**C**) GSE42955, (**D**) GSE43435, (**E**) GSE79962, (**F**) GSE84796.

**Figure 2 ijms-24-07339-f002:**
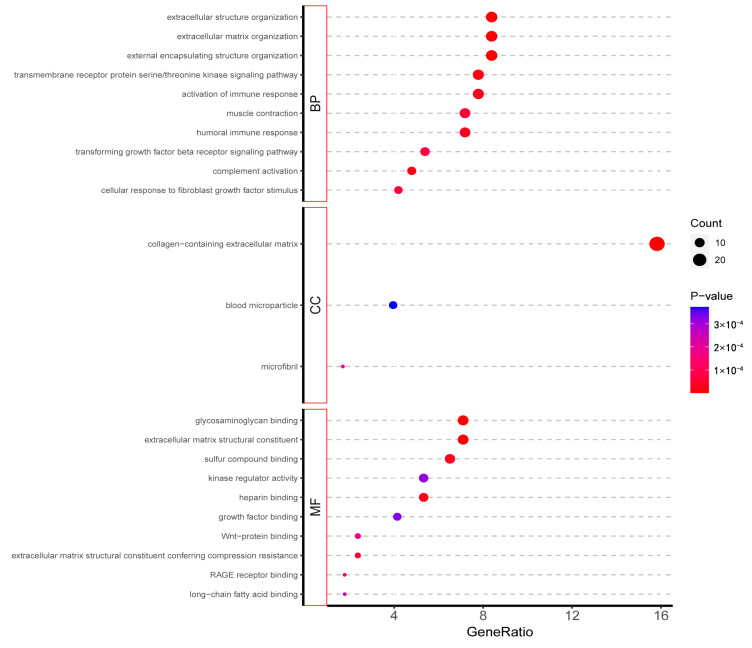
GO enrichment analysis results of RRA-analysed differential genes.

**Figure 3 ijms-24-07339-f003:**
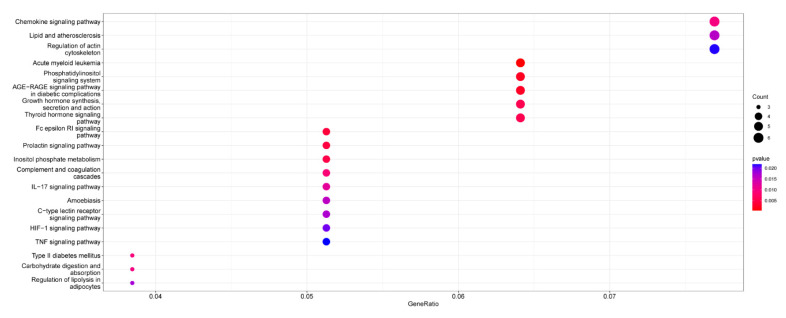
KEGG pathway enrichment analysis of RRA-analysed differential genes.

**Figure 4 ijms-24-07339-f004:**
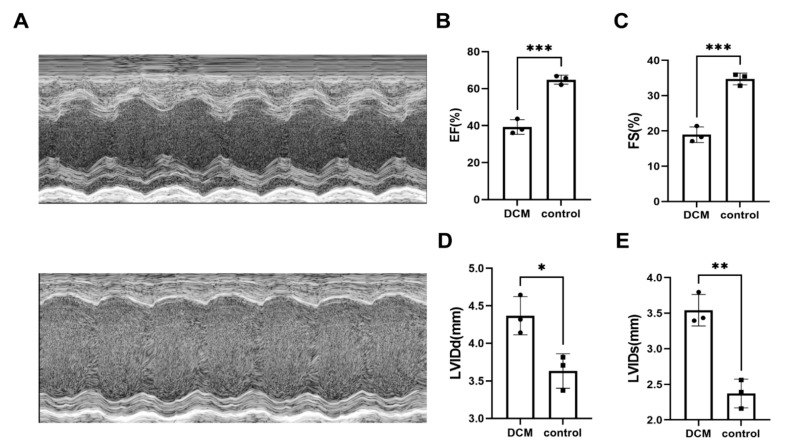
Representative echocardiograms (**A**) and related cardiac function indices (**B**–**E**) of healthy and DCM mice. * represent *p*-value < 0.05; ** represent *p*-value < 0.01; *** represent *p*-value < 0.001.

**Figure 5 ijms-24-07339-f005:**
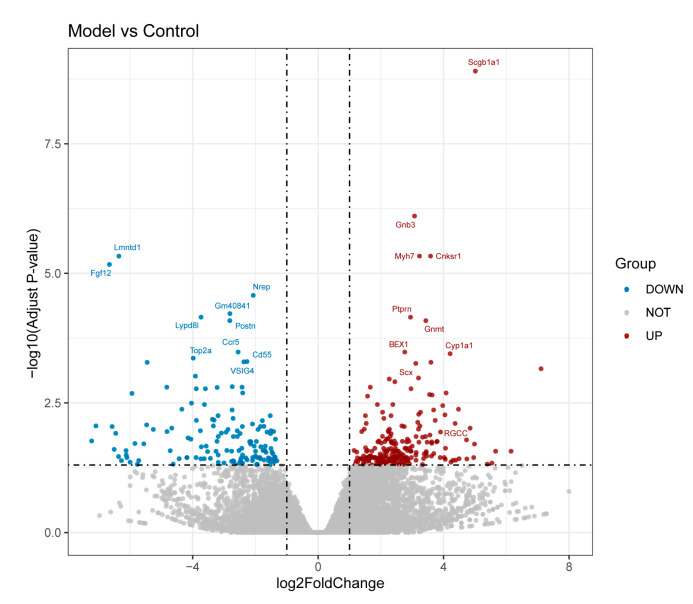
Volcano diagram of RNA-Seq results. Red dots indicate upregulated genes, blue dots downregulated genes, and grey dots indicate genes with no significant change.

**Figure 6 ijms-24-07339-f006:**
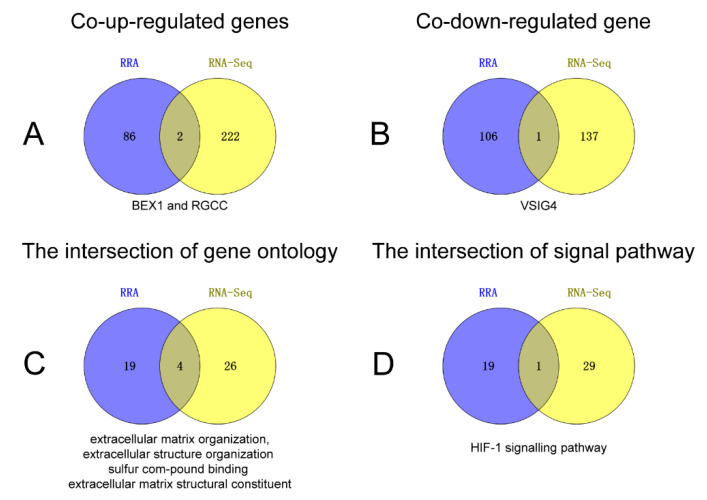
The Venn diagram indicating the intersection of RRA and RNA-Seq analysis results. (**A**) Upregulated genes, (**B**) downregulated gene, (**C**) the intersection of GO, and (**D**) the intersection of KEGG.

**Figure 7 ijms-24-07339-f007:**
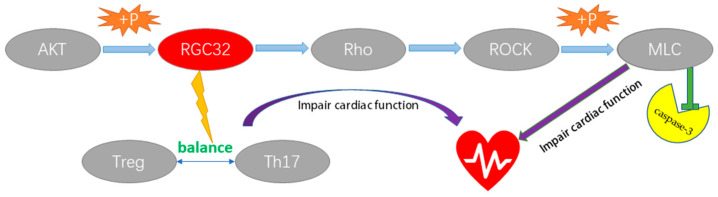
RGC32 may impair cardiac function through the Rho/ROCK axis and immune cell pathways.

**Figure 8 ijms-24-07339-f008:**
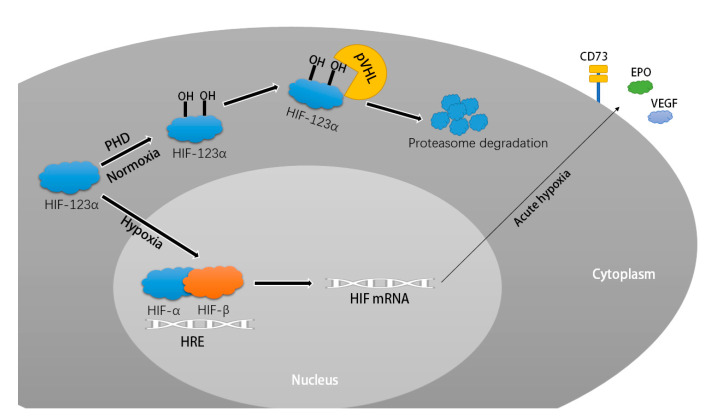
Degradation of HIF under normoxic conditions and the HIF signalling pathway under acute ischaemic conditions mediates protection through VEGF, EPO, and CD73.

**Table 1 ijms-24-07339-t001:** Details of microarray analysis.

GSE ID	Participants	Platform	Year	Number of DEGs		Reference
				Up	Down	
GSE3585	7 cases and 5 controls	GPL96	2005	147	123	Barth et al. [7]
GSE29819	7 cases and 6 controls	GPL570	2011	1132	1103	Gaertner et al. [8]
GSE42955	12 cases and 5 controls	GPL6244	2012	79	102	Molina-Navarro et al. [9]
GSE43435	10 cases and 10 controls	GPL15338	2013	365	518	Koczor et al. [10]
GSE79962	9 cases and 11 controls	GPL6244	2016	349	252	Matkovich et al. [11]
GSE84796	10 cases and 7 controls	GPL14550	2016	3223	4022	Laugier et al. [12]

**Table 2 ijms-24-07339-t002:** Binary logic regression results.

Term	Correlation Coefficient	SE	Wald	*p*	OR	95% Confidence Interval of OR	
						Upper Limit	Lower Limit
*BEX1*	1.513	0.528	8.197	0.004 ***	4.538	1.611	12.782
*RGCC*	1.005	0.51	3.882	0.049 **	2.731	1.005	7.42
*VSIG4*	1.102	0.509	4.692	0.030 **	3.009	1.11	8.152

Note: *** and ** represent the significance level of 1% and 5%, respectively.

## Data Availability

The data of RNA-Seq in this publication have been deposited in the NCBI Gene Expression Omnibus and are accessible through GEO Series accession number GSE224211 (https://www.ncbi.nlm.nih.gov/geo/query/acc.cgi?acc=GSE224211 (accessed on 11 April 2023)).

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
