# Peer review of "Identification of Signature Genes of Dilated Cardiomyopathy Using Integrated Bioinformatics Analysis"

_ijms, 2023, doi:10.3390/ijms24087339_

Round 1

Reviewer 1 Report (Previous Reviewer 2)

Thank you to the authors for the revision. I appreciate the concluding statement to the abstract. The manuscript reads well, and additional analyses and the venn diagrams highlighting the intersections between RRA and RNA-seq are a nice addition. 

Reviewer 2 Report (New Reviewer)

The research manuscript “Identification of Signature Genes of Dilated Cardiomyopathy Using Integrated Bioinformatics Analysis” by Wu et al identifies molecular changes associated with the development of Dilated cardiomyopathy (DCM) using publicly available datasets from GEO database and experiments on DCM animal model. Authors have identified three genes of high significance including Bex1, RGCC and VSIG4 out of all other differential expressed genes. Significantly deregulated these three genes in DCM were analyzed using binary logistic regression analysis.

Authors have also highlighted key biological processes like extracellular matrix organization, extracellular structural organization, and HIF-1 signaling pathway associated with the differently expressed genes. The study can serve as a platform for future studies performing more detailed analysis using patient sample data and bigger sample size of animal studies.

The study can be accepted in its current form as a preliminary finding.

This manuscript is a resubmission of an earlier submission. The following is a list of the peer review reports and author responses from that submission.

Round 1

Reviewer 1 Report

In this manuscript the authors describe a study that combine the public database resources and RNA-Seq technology to explore and to identify the genes responsible for dilated cardiomyopathy. Through this comprehensive analysis, they have identified three key differential genes associated with dilated cardiomyopathy.

The manuscript is well written, and the work is well organized. The limitation of this study is that the genes identified as altered in the pathology should at least be verified in the animal model, otherwise the study remains only speculative.

Minor points:

The GEO data references of GSE3585, GSE29819, GSE42955, GSE43435, GSE79962, and GSE84796 should be cited in the manuscript.

In Table 1 the "Tissues” and “Analysis type" columns should be eliminated and a final column should be added which shows the references of the GEO data.

Reviewer 2 Report

In the manuscript titled “Identification of signature genes od Dilated cardiomyopathy using integrated bioinformatics analysis”, Zhimin Wu and team sought to key genes associated with the pathogenesis of dilated cardiomyopathy (DCM) by use of public databases and RNA-seq. The manuscript is well-written and highlights some differentially expressed genes (DEGs) associated with DCM, however I note my concerns below.

Major concerns:

1.      What is the primary purpose of the manuscript? Is it to share the approach of finding DEGs with microarray data, RRA method and RNA seq, or to share the science you discovered such as genes of interest and the possible biology of their roles in DCM pathophysiology? If it’s the latter, I suggest tailoring some key plots and writing to highlight the DEGs/enrichment pathways and their potential roles in DCM, and how the methods you used to help unravel the key findings.

2.      How are the plots and figures adding to the manuscript? I suggest moving Table 1 and 2 to supplementary materials. For Figure 1, you could also move to supplementary material, or you can label some DEGs on the volcano plots in the different microarrays.

3.      Please add the DEGs on the volcano plot in Figure 5.

4.      Why was RNA-seq performed on the doxorubicin-induced mouse models in addition to the microarray data? How were the findings integrated or validated?

Minor concerns:

1.      Abstract—please define RRA.

2.      In the Introduction section, you shared some genes that were found to be associated with DCM (TTN, MYH7 and MYH6)—I would mention whether these were among the DEGs found in your results. If not, expand on why this might have been the case (for ex: earlier studies had lower sample size, etc).

3.      Is it possible to add the number of genes detected in each of the array datasets in Table 1?

4.      Can you expand on the GO and KEGG analyses findings in Discussion (as in how these may apply to DCM)?

Round 2

Reviewer 1 Report

The authors have corrected the manuscript according to requests, so I believe it can be accepted for publication.

Author Response

We sincerely thank you for your approval of our manuscript. It is your valuable opinions that make our manuscript more scientific and reasonable.

Reviewer 2 Report

Thank you to the authors for the extensive amount of work and clarifications in this version of the manuscript titled, "Identification of signature genes of Dilated cardiomyopathy 2 Using Integrated bioinformatics analysis". It reads much better, the figures are informative and the diagrams highlighting the potential mechanics of DCM are appreciated. I only have a few minor concerns to add.

1. Please add a short conclusion or a finishing statement to the Abstract

2. There are a few grammatical issues-- please review (for ex} line 127 of the manuscript is incomplete; need to be consistent in using "upregulated/downregulated" or "up-regulated/down-regulated").

3. Thank you for the clarification on RRA. I only meant that the prior manuscript did not have what RRA was an acronym for. This is now resolved. 
